# Usefulness of Muscle Ultrasonography in the Nutritional Assessment of Adult Patients with Cystic Fibrosis

**DOI:** 10.3390/nu14163377

**Published:** 2022-08-17

**Authors:** Francisco José Sánchez-Torralvo, Nuria Porras, Ignacio Ruiz-García, Cristina Maldonado-Araque, María García-Olivares, María Victoria Girón, Montserrat Gonzalo-Marín, Casilda Olveira, Gabriel Olveira

**Affiliations:** 1Unidad de Gestión Clínica de Endocrinología y Nutrición, Hospital Regional Universitario de Málaga, 29007 Malaga, Spain; 2Instituto de Investigación Biomédica de Málaga (IBIMA), Plataforma Bionand, 29010 Malaga, Spain; 3Departamento de Medicina y Dermatología, Facultad de Medicina, University of Malaga, 29010 Malaga, Spain; 4Unidad de Gestión Clínica de Neumología, Hospital Regional Universitario de Málaga, 29010 Malaga, Spain; 5Centro de Investigación Biomédica en Red de Diabetes y Enfermedades Metabólicas Asociadas (CIBERDEM), Instituto de Salud Carlos III, 28029 Madrid, Spain

**Keywords:** ultrasonography, muscle ultrasound, muscle mass, cystic fibrosis, malnutrition, GLIM criteria

## Abstract

Background: Muscle ultrasonography of the quadriceps rectus femoris (QRF) is a technique on the rise in the assessment of muscle mass in application of nutritional assessment. The aim of the present study is to assess the usefulness of muscle ultrasonography in patients with cystic fibrosis, comparing the results with other body composition techniques such as anthropometry, bioelectrical impedance analysis (BIA), dual-energy X-ray absorptiometry (DXA), and handgrip strength (HGS). At the same time, we intend to assess the possible association with the nutritional and respiratory status. Methods: This was a prospective observational study in adult patients with cystic fibrosis in a clinically stable situation. Muscle ultrasonography of the QRF was performed, and the results were compared with other measures of body composition: anthropometry, BIA, and DXA. HGS was used to assess muscle function. Respiratory parameters were collected, and nutritional status was assessed using Global Leadership Initiative on Malnutrition (GLIM) criteria. Results: A total of 48 patients were included, with a mean age of 34.1 ± 8.8 years. In total, 24 patients were men, and 24 patients were women. Mean BMI was 22.5 ± 3.8 kg/m^2^. Mean muscular area rectus anterior (MARA) was 4.09 ± 1.5 cm^2^, and mean muscular circumference rectus was 8.86 ± 1.61 cm. A positive correlation was observed between the MARA and fat-free mass index (FFMI) determined by anthropometry (*r* = 0.747; *p* < 0.001), BIA (*r* = 0.780; *p* < 0.001), and DXA (*r* = 0.678; *p* < 0.001), as well as muscle function (HGS: *r* = 0.790; *p* < 0.001) and respiratory parameters (FEV1; *r* = 0.445, *p* = 0.005; FVC: *r* = 0.376, *p* = 0.02; FEV1/FVC: *r* = 0.344, *p* = 0.037). A total of 25 patients (52.1%) were diagnosed with malnutrition according to GLIM criteria. Differences were observed when comparing the MARA based on the diagnosis of malnutrition (4.75 ± 1.65 cm^2^ in normo-nourished vs. 3.37 ± 1.04 in malnourished; *p* = 0.014). Conclusions: In adults with cystic fibrosis, the measurements collected by muscle ultrasound of the QRF correlate adequately with body composition techniques such as anthropometry, BIA, DXA, and handgrip strength. Muscle ultrasound measurements, particularly the MARA, are related to the nutritional status and respiratory function of these patients.

## 1. Introduction

Cystic fibrosis (CF) is a disease caused by the alteration of a single gene located on the long arm of chromosome 7—the CFTR gene (regulator of the transmembrane conductance of cystic fibrosis). It consists of a multisystemic disease in which CFTR protein dysfunction causes alteration of ion transport in the apical membrane of epithelial cells in different organs and tissues. It is the most frequent life-threatening disease, with recessive Mendelian inheritance, in the Caucasian population [1].

Until a few years ago, CF was considered to be associated with malnutrition because it was practically always present at the time of diagnosis, and because the vast majority of patients suffered a deterioration in their nutritional status and died severely malnourished. Currently, the prevalence of malnutrition has decreased notably, although figures close to 25% continue to be reported in both children and adults [2], reaching up to 40% depending on the tool used [3]. 

Malnutrition affects the respiratory muscles and, thus, lung function, decreases exercise tolerance, and leads to immunological alteration. In both children and adults, it behaves as a predictive risk factor for morbidity and mortality and deterioration in quality of life [4,5,6,7,8]. In relation to lung function, there seems to be a clear interrelation between malnutrition and its deterioration, as well as chronic colonization (especially by *Pseudomonas*), which is accentuated with age. This is why nutritional intervention can, in addition to improving nutritional parameters, slow down the progressive decline in lung function.

Although, until recently, the recommendations focused mainly on achieving adequate weight, height, and body mass index (BMI) [9], the need to also assess fat-free mass and its functionality is currently highlighted, since its decrease is associated with a worse prognosis [4,5,7,8]. In fact, there is an increasing prevalence of overweight and obesity in CF, particularly in the era of modulator therapies [10]. Nevertheless, it is possible to be obese or overweight and have a depletion in fat-free mass. The decrease in fat-free mass has been associated with an increase in systemic inflammation that is observed in these patients [11].

Therefore, in patients with cystic fibrosis, it is recommended to collect body composition measurements [5,12]. To do this, different tools are available. The simplest and cheapest techniques would be the measurement of skinfolds and circumferences (at least the triceps skinfold and brachial circumference) and bioelectrical impedance analysis (BIA). Both techniques have been validated in CF [13,14]. Other useful methods to measure body compartments (less used due to their greater complexity and cost) are dual-energy X-ray absorptiometry (DXA), which could be considered the “gold standard” available to clinicians (also validated in CF), and other more complex ones such as computerized tomography (CT), magnetic resonance (MR), and the doubly labeled water technique [5,15,16,17,18]. 

In our series of adults with cystic fibrosis, we have been carrying out some of these body composition assessment techniques. Using BIA, 16.2% of our patients presented a low fat-free mass, compared to 17.7% using anthropometry and increasing to 37.6% using DXA [19]. 

In 2018, Global Leadership Initiative on Malnutrition (GLIM) criteria [20] were proposed for the diagnosis of malnutrition. These criteria include as a novelty the evaluation of muscle mass, proposing the use of the aforementioned techniques. However, there are other techniques for assessing muscle mass, both directly and indirectly, which are not presented as the first choice in the consensus. One of the techniques not originally included in the GLIM criteria is muscle ultrasonography of the quadriceps rectum (QRF), although its recommendation is recently beginning to be assessed [21]. This technique has been used since the 1980s to assess muscle status, but its use for diagnosing malnutrition is not as widespread. There are studies suggesting that this technique is reliable and less susceptible to errors due to swelling or inflammation than other techniques used in clinical practice such as BIA [22,23], in addition to having as an advantage over DXA or CT the absence of radiation [22]. In short, it is a simple, bloodless, and accessible technique to assess body composition in patients at risk of malnutrition or malnourished. And its use is on the rise [23,24], although there are presently no cutoff points described for the diagnosis of malnutrition, or its prognostic value in relation to complications has not been described [25]. Muscle ultrasonography has previously been used to assess muscle mass in children with cystic fibrosis [26], but there is currently no study that evaluated its use in the adult population.

Taking into consideration the above, the diagnosis and prevention of malnutrition are essential in patients with CF, paying special attention to muscular body composition, for which a simple, accessible, and reliable method is needed that can be performed as a routine analysis in consultation, such as muscle ultrasound of the QRF [27].

Our hypothesis is that muscle ultrasonography could be an appropriate technique for measuring muscle mass in the nutritional assessment of patients with cystic fibrosis, and that it would have good agreement with other diagnostic tools for estimating muscle mass and strength.

Thus, the aim of the study was to assess the usefulness of muscle ultrasonography in the diagnosis of malnutrition in patients with cystic fibrosis, comparing the results with other body composition techniques such as anthropometry, handgrip strength, BIA, and DXA. At the same time, we intend to assess the possible association with the nutritional and respiratory status during patient follow-up.

## 2. Materials and Methods

We designed a prospective observational study of routine clinical practice. Adult patients with a diagnosis of cystic fibrosis in a situation of clinical stability assessed at the Nutrition Unit of the UGC of Endocrinology and Nutrition of the Hospital Regional Universitario de Malaga were selected, coinciding with the annual study that is usually carried out.

### 2.1. Anthropometric and Body Composition Parameters

Weight was assessed through BIA (scale mode, weight function; TANITA MC980MA) and height was obtained using a stadiometer (Holtain limited, Crymych, UK). With these two values, BMI was calculated. 

BIA was performed with TANITA MC980MA (TANITA Corporation, Tokyo, Japan), providing information about total body composition (phase angle and fat-free mass).

Dual-energy X-ray absorptiometry (DXA) was performed using a Lunar Prodigy Advance densitometer (General Electric Medical Systems). Fat-free mass was recorded. The software used was EnCore 12.3 (iDXA and Prodigy Advance). 

The skinfolds assessed were the triceps, biceps, subscapularis, and supra-iliac using a Holtain constant pressure caliper (Holtain Limited, Crymych, UK). The same investigator (N.P.) performed the measurements in triplicate for each of the skinfolds assessed, and the mean was calculated. Fat mass and fat-free mass (FFM) were estimated according to the formulas of Siri and Durnin [28,29]. Age, sex, weight, and the sum of four skinfolds (triceps, biceps, supra-iliac, and subscapular) were taken into account in the formula. 

The fat-free mass index (FFMI) was calculated for anthropometry, BIA, and DXA.

Muscle strength was assessed using a Jamar dynamometer (Asimow Engineering Co., Los Angeles, CA, USA) and was performed in the dominant hand, repeated three times. The mean was calculated.

### 2.2. Muscle Ultrasonography of the Quadriceps Rectus Femoris (QRF)

Muscle ultrasonography of the QRF was performed on the nondominant lower extremity with the patient in a supine position, using a 10–12 MHz probe and a multifrequency linear matrix color Esaote MyLab Gamma (Esaote, Genova, Italy). The probe was aligned perpendicular to the longitudinal and transverse axis of the nondominant QRF. The evaluation was performed without compression at the level of the lower third from the superior pole of the patella and the anterior superior iliac spine, measuring the *X*- and *Y*-axis (transverse muscle thickness), circumference (CMR), cross-sectional area (MARA), and transverse subcutaneous adipose tissue (cm) (SCAT) [30]. The index of the muscle to height (cm^2^/m^2^) (MARAI) was calculated. The ultrasonography was performed by the same person (G.O.) who was familiar with the technique. Three measurements were performed for each parameter, and the mean was calculated. 

### 2.3. Assessment of Nutritional Status

A nutritional assessment was performed according to GLIM criteria. For achieving a diagnosis of malnutrition, the presence of at least one phenotypic criterion and one etiologic criterion was required [20]. The following phenotypic criteria were evaluated: more than 5% unintentional weight loss in 6 months, a low BMI (BMI below 20 kg/m^2^), or a decrease in muscle mass determined by DXA (gold standard). Regarding etiological criteria, reduced intake and reduced assimilation were assessed, especially the existence of pancreatic insufficiency. Lastly, chronic disease-related inflammation was assessed using the Glasgow Prognostic Score [31].

### 2.4. Assessment of Respiratory Status

The exacerbations recorded during the annual examination were assessed, taking into consideration those happening in the year prior to the evaluation. Such exacerbations were classified into mild/moderate or severe (suggestive symptoms that worsen and require hospitalization and/or intravenous antibiotics on an outpatient basis). Moreover, patients underwent forced spirometry using a JAEGER pneumotachograph (Jaeger Oxycon Pro^®^, Erich Jaeger, Würzberg, Germany), following the Sociedad Española de Neumología y Cirugía Torácica (SEPAR) guidelines and determining the values of forced vital capacity (FVC), forced expiratory volume in 1 s (FEV1), and the ratio between both (FEV1/FVC). The values were expressed in absolute terms in mL and as percentages according to a reference population [32]. Initial colonization by microorganisms was analyzed, taking into account their appearance in sputum (at least three positive occurrences), regardless of their persistence at the time of the study. Measurement of the mean amount of sputum produced daily (in milliliters) was evaluated following the protocol of Martínez-García et al. [33].

### 2.5. Statistical Analysis

Quantitative variables were expressed as the mean ± standard deviation. The distribution of quantitative variables was assessed using the Kolmogorov–Smirnov test. Differences between quantitative variables were analyzed using Student’s *t*-test and, for variables not following a normal distribution, using nonparametric tests (Mann–Whitney). The associations of the variables were evaluated by estimating the Pearson or Spearman correlation coefficient, according to normality. For calculations, significance was set at *p* < 0.05 for two tails.

Evaluations of the diagnostic performance of muscle ultrasound variables to detect malnutrition according to GLIM criteria were based on the receiver operating characteristic (ROC) curves and the area under the curve (AUC). We estimated the accuracy of these measurements using AUC by plotting sensitivity versus 1 − specifity. ROC curves were used to determine the optimal cutoff values by finding the point maximizing the product of sensitivity and specificity. Data analysis was performed using the JAMOVI software (version 2.2.2, Jamovi project, 2020). 

### 2.6. Ethics

All subjects gave their informed consent for inclusion before they participated in the study. The study was conducted in accordance with the Declaration of Helsinki, and the protocol was approved by the Research Ethics Committee of Malaga on 30 March 2021 (reference number #30032021).

## 3. Results

A total of 48 patients were included. They had a mean age of 34.1 ± 8.8 years. Of the sample, a total of 24 were men (50%) and 24 were women (50%). Half of the individuals were heterozygous for ΔF508, and 77% had pancreatic insufficiency. Furthermore, 18 patients (37.5%) presented disease-related inflammation (Glasgow prognostic score > 0).

According to the BMI, only 8.3% of women and 12.5% of men had malnutrition (BMI less than 18.5 kg/m^2^). In the application of the GLIM criteria for the diagnosis of malnutrition, we found that seven patients (14.5%) had lost more than 5% of their weight in the previous 6 months, and 13 patients (27%) had a BMI below 20 kg/m^2^.

Using DXA as a determinant of muscle mass, 20 patients (41.7%) had an FFMI below the cutoff points (61.1% of women and 23.5% of men). With these data, 25 patients (52.1%) were diagnosed with malnutrition according to GLIM criteria (64% of women and 36% of men).

Table 1 shows the general characteristics and respiratory status of the sample, as well as its adjustment for nutritional status.

### 3.1. Body Composition and Other Anthropometric Measurements

Table 2 shows the values of the morphofunctional study performed on the subjects, including anthropometry, BIA, DXA, handgrip strength, and the results of the muscle ultrasound parameters.

### 3.2. Correlation between Ultrasound and Other Anthropometric and Body Composition Measurements

Table 3 shows the correlation between the muscle ultrasound parameters and other morphofunctional parameters.

The correlation between the muscle ultrasonography measurements and BMI, although statistically significant, was weak.

A statistically significant correlation was found between the fat-free mass determined by anthropometry and the muscle ultrasound variables, highlighting a good correlation with the muscular area rectus anterior (MARA) (*r* = 0.747; *p* < 0.001, Figure 1a) and rectus muscular circumference (*r* = 0.736; *p* < 0.001, Figure 1b). Likewise, a negative correlation was observed between subcutaneous adipose tissue and fat-free mass, a finding that also occurred in the remaining morphofunctional parameters (Table 3).

Regarding the BIA measurements, statistically significant correlation was found between the fat-free mass determined by anthropometry and the muscle ultrasound variables, highlighting a good correlation with the muscular area rectus anterior (MARA) (*r* = 0.780; *p* < 0.001, Figure 1c) and rectus muscular circumference (*r* = 0.763; *p* < 0.001, Figure 1d) (Table 3).

Fat-free mass determined by DXA also achieved a statistically significant positive correlation with ultrasound measurements, as did MARA (*r* = 0.670; *p* < 0.001, Figure 1e) and muscular circumference (*r* = 0.677; *p* < 0.001, Figure 1f) (Table 3).

### 3.3. Correlation between Ultrasound and Muscle Strength

Handgrip strength showed a good correlation with the ultrasound measurements of MARA (*r* = 0.790; *p* < 0.001; Figure 2a) and muscle circumference (*r* = 0.779; *p* < 0.001; Figure 2b).

### 3.4. Respiratory Variables

Table 4 shows the correlations between the ultrasound measurements and the different respiratory variables.

### 3.5. Nutritional Status

A comparison between muscle ultrasound parameters according to the nutritional status of the patient is shown in Table 5. Significant differences were found between normo-nourished and malnourished patients in the measures of MARA (4.75 ± 1.65 cm^2^ vs. 3.37 ± 1.04 cm^2^, *p* = 0.014) and *Y*-axis (1.45 ± 0.36 cm vs. 1.17 ± 0.28, *p* = 0.010).

Using the ROC curve, we determined the MARA cutoff points for predicting malnutrition (Figure 3). ROC curve analysis showed that MARA had a significant discriminative ability to detect malnutrition. The MARA cutoff for malnutrition diagnosis was 2.97 cm^2^, with AUC = 0.664 (sensitivity 72.7% and specificity 69.2%), in women and 4.71 cm^2^, with AUC = 0.732 (sensitivity 68.8% and specificity 85.7%), in men. 

## 4. Discussion

To our knowledge, this is the first study to include the assessment of muscle ultrasonography of QRF in adult patients with cystic fibrosis. We found that muscle ultrasonography correlates with handgrip strength and body composition measurement techniques such as BIA, DXA, and anthropometry in patients with cystic fibrosis. In turn, we found an association between this tool and nutritional status and respiratory parameters.

Malnutrition and muscle mass depletion are related to an increase in complications and a worse prognosis in patients with cystic fibrosis. To assess the nutritional status of patients with cystic fibrosis, it is necessary to perform measurements of body composition beyond BMI. We used muscle ultrasonography to assess body composition as it is a simple, accessible technique, and free of radiation. Our study found a good correlation between muscle ultrasound measurements and the remaining morphofunctional assessment techniques, consolidating the results of previous studies [35]. As expected, there were differences in the ultrasound measurements of men and women in all the parameters measured.

Previous studies found that the prevalence of low FFM is high in adult CF patients, despite a normal BMI [3]. In our study, we observed a low correlation of the BMI with the ultrasound parameters. BMI was especially related to the amount of body fat mass. We also found an association of a similar magnitude with the subcutaneous adipose tissue of the leg, which suggests that this isolated measure does not adequately define the amount of total body fat.

However, muscle ultrasound measurements did show a good correlation with fat-free mass, using both BIA and anthropometry. The ultrasound parameters that best correlated with estimated FFM were MARA and muscle circumference. On the other hand, an association was also found between ultrasound and phase angle, a biomarker for malnutrition, hydration, and inflammatory status [36]. Muscle ultrasound could, therefore, be a reliable technique that helps to quantify the active cell mass, which could justify the evolutionary monitoring of the nutritional status in patients with cystic fibrosis together with the other morphofunctional assessment techniques.

Although DXA is considered the “gold standard” in the clinic for estimating fat-free mass, the ultrasound parameters showed slightly lower correlations than those observed with the other techniques. Other previous works have pointed out differences in the estimation of fat-free mass between DXA and BIA and anthropometry in patients with CF [37], finding some overestimation in FFM using skinfold measurements and BIA [38]. Similar findings have been reported in patients with bronchiectasis [39]. 

Handgrip strength is a muscle assessment technique that is a good functional and global health status marker. It is a good marker of renutrition [40], and it has been shown that it can be used as a determinant of muscle mass in the application of the GLIM criteria [41]. In our study, we found a strong correlation with the muscle ultrasound measurements, even higher than with the techniques described above. These findings would support the claim that handgrip strength is a good estimator of muscle mass [42], despite not being recommended for muscle assessment according to the last guidelines [21]. In our experience, handgrip strength is associated with the nutritional status and respiratory parameters (lung function and exacerbations) in patients with CF [43], results similar to those found with muscle ultrasound in the present study. This fact postulates muscle ultrasound, as well as HGS, as a good predictor of the nutritional and respiratory status of patients with cystic fibrosis.

In our study, we found an association between muscle ultrasound parameters (especially the MARA and *Y*-axis) and respiratory function parameters. Even without being able to establish causal relationships, patients with lower muscle mass determined by ultrasound seemed to have worse FEV1, FVC, and FEV1/FVC values. This finding was previously described in patients with low muscle strength [44,45] or poor results in anthropometry, BIA, and DXA [8,15,40,46]. In this case, muscle ultrasonography is postulated as a good predictor of respiratory function, although further studies are needed.

To our knowledge, this is the first study to show the prevalence of malnutrition in adult patients with CF applying the GLIM criteria. Malnourished patients presented worse muscle ultrasound values, again best represented on the MARA and *Y*-axis. In previous studies, it was reported that malnourished patients with CF presented worse handgrip strength values [43]. Regarding the ability of the MARA to establish cutoff points for malnutrition according to GLIM criteria, it should be noted that the cutoff points offered by the ROC curve were close to the MARA means obtained in our measurements. This could be justified because almost half of the patients were malnourished according to GLIM criteria. This does not coincide with the prevalence previously reported [2], where muscle mass was not included. In our previous series, the prevalence of low fat-free mass was high (almost 50% especially in women) in adult patients with CF, despite having a normal BMI [3]. Therefore, the use of malnutrition diagnostic tools that include measurement of muscle mass is especially relevant in these patients. 

This study had some limitations. It was a cross-sectional study, which prevented us from extracting causal conclusions; thus, we could only speculate on different associations. Moreover, it was a single-center study on a moderate number of participants. On the other hand, the muscle ultrasound measurement technique is not yet universally accepted, as nutritional ultrasound is considered to be a developing technique, lacking population reference values and widespread cutoff points [35].

Nonetheless, as strengths of our study, we highlight that all measurements were performed by a single, experienced observer using standardized methodology [30]. Furthermore, several body composition techniques were used, from the most accessible to the gold standard in clinical use, which enhances the reliability of the results and brings it closer to clinical practice.

In conclusion, in adults with cystic fibrosis, the measurements collected by muscle ultrasonography of the quadriceps rectus femoris correlate adequately with body composition techniques such as anthropometry, BIA, DXA, and handgrip strength. Muscle ultrasound measurements, particularly the MARA and *Y*-axis, are related to the nutritional status and respiratory function of these patients. It is a simple, accessible, reliable, low-cost, and noninvasive technique that could be used to assess the muscle mass of these patients, helping to diagnose malnutrition and monitor the evolution in patients with CF. However, further studies are required to provide information on normal values and widespread cutoff points.

## Figures and Tables

**Figure 1 nutrients-14-03377-f001:**
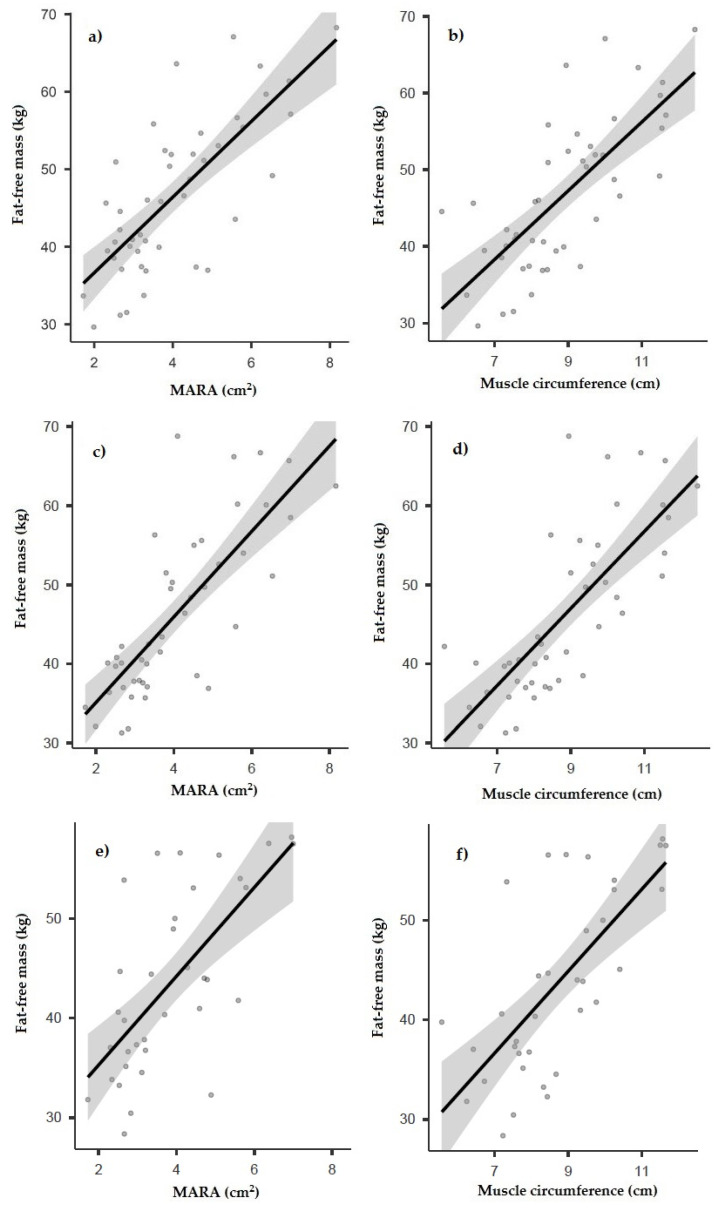
Correlation between MARA and muscle circumference and fat-free mass determined by BIA (**a**,**b**), anthropometry (**c**,**d**), and DXA (**e**,**f**). MARA: muscular area rectus anterior.

**Figure 2 nutrients-14-03377-f002:**
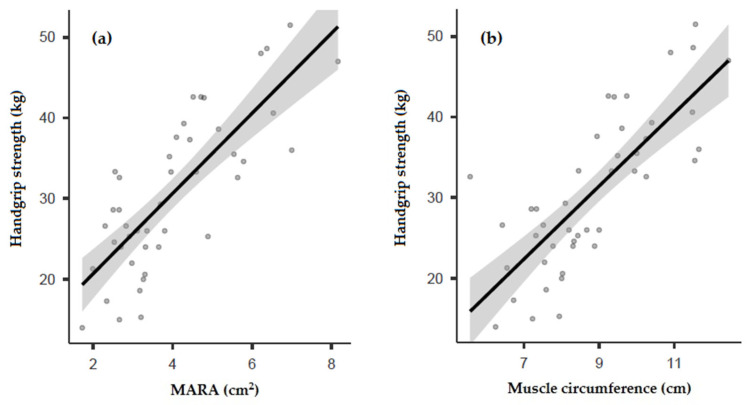
Correlation of handgrip strength with MARA (**a**) and muscle circumference (**b**). MARA: muscular area rectus anterior.

**Figure 3 nutrients-14-03377-f003:**
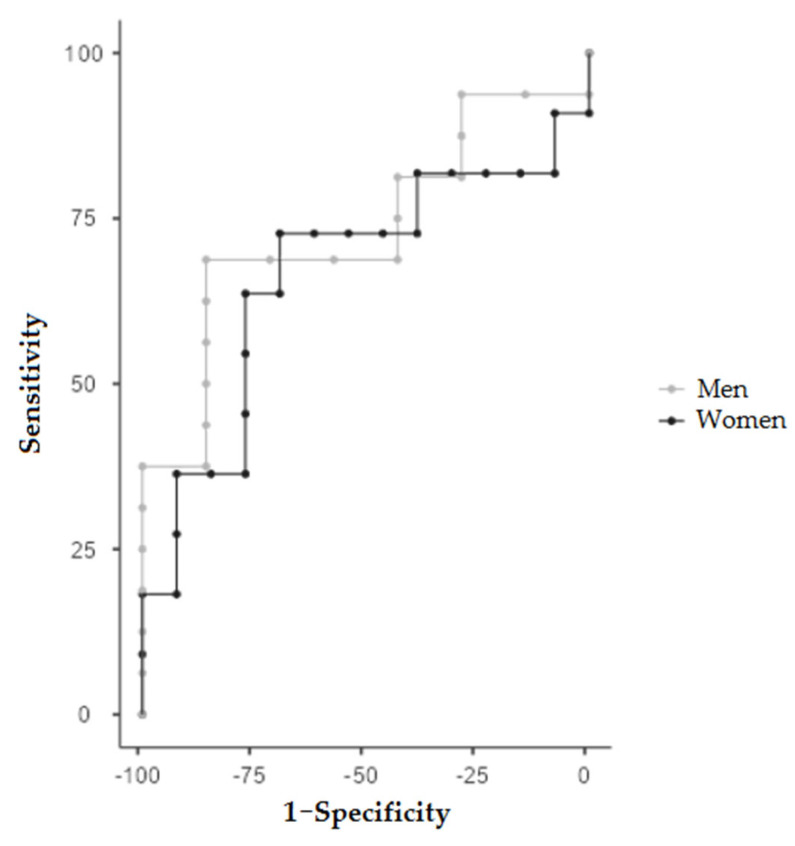
ROC curve analyses for muscular area rectus anterior (MARA) to detect malnutrition.

**Table 1 nutrients-14-03377-t001:** General characteristics and respiratory status, adjusted by nutritional status.

		Overall	Normo-Nourished	Malnourished	*p*-Value
		*n* = 48	*n* = 23	*n* = 25	
Age (years)	*m* ± SD	34.1 ± 8.8	35 ± 8.8	32.9 ± 8.9	0.43
Gender	*n* (%)				
Men		24 (50)	15 (62.2)	9 (36)	
Women		24 (50)	8 (34.8)	16 (64)	0.02
Mutation	*n* (%)				
Homozygous for ΔF508		11 (22.9)	6 (26.1)	5 (20)	
Heterozygous for ΔF508		24 (50)	9 (39.1)	15 (60)	
Negative for ΔF508		13 (27.1)	8 (34.8)	5 (20)	0.26
Cystic fibrosis-related diabetes [34]	*n* (%)	25 (52.1)	11 (47.8)	14 (56)	0.68
Pancreatic insufficiency	*n* (%)	37 (77.1)	17 (73.9)	20 (80)	0.76
Bronchorrhea (mL)	*m* ± SD	22 ± 22.5	20.5 ± 24.1	22.3 ± 20.3	0.81
Total exacerbations	*m* ± SD	0.63 ± 0.93	0.73 ± 1.03	0.54 ± 0.83	0.51
Severe exacerbations	*m* ± SD	0.13 ± 0.4	0.18 ± 0.5	0.08 ± 0.28	0.41
FEV 1 (%)	*m* ± SD	59.4 ± 24.1	65.7 ± 22.6	52.5 ± 24.5	0.09
FVC (%)	*m* ± SD	67.6 ± 19.8	70.3 ± 18.4	64.5 ± 21.3	0.38
FEV1/FVC (%)	*m* ± SD	0.69 ± 0.11	0.73 ± 0.09	0.64 ± 0.11	0.01
Colonizations	*n* (%)	41 (87.2)	21 (91.3)	20 (80)	0.13
*Pseudomonas aeruginosa*		37 (77.1)	19 (82.6)	18 (72)	0.14
*Staphylococcus aureus*		38 (79.2)	20 (86.9)	18 (72)	0.10
*Haemophilus influenzae*		23 (47.9)	12 (52.2)	11 (44)	0.47

*m*: mean; SD: standard deviation; FEV1: forced expiratory volume in 1 s; FVC: forced vital capacity.

**Table 2 nutrients-14-03377-t002:** Body composition parameters.

		Men (*n* = 24)	Women (*n* = 24)	*p*-Value
BMI (kg/m^2^)	Mean ± SD	23.2 ± 4.2	21.8 ± 3.4	0.21
Triceps skinfold (mm)	Mean ± SD	9.26 ± 4.4	17.3 ± 5.6	*p* < 0.001
Arm muscle circumference (cm)	Mean ± SD	24.8 ± 2.4	20.5 ± 2.4	*p* < 0.001
Fat-free mass (anthropometry) (kg)	Mean ± SD	54.1 ± 7.1	39.1 ± 5.8	*p* < 0.001
FFMI (anthropometry) (kg/m^2^)	Mean ± SD	18.9 ± 1.9	15.6 ± 1.4	*p* < 0.001
Phase angle (°)	Mean ± SD	6.07 ± 0.6	5.02 ± 0.54	*p* < 0.001
Fat-free mass (BIA) (kg)	Mean ± SD	54.4 ± 8.1	38.6 ± 5.7	*p* < 0.001
FFMI (BIA) (kg/m^2^)	Mean ± SD	18.9 ± 2.2	15.4 ± 1.5	*p* < 0.001
Fat-free mass (DXA) (kg)	Mean ± SD	51.8 ± 6	37.3 ± 6	*p* < 0.001
FFMI (DXA) (kg/m^2^)	Mean ± SD	18.2 ± 1.4	14.6 ± 1.6	*p* < 0.001
Handgrip strength (kg)	Mean ± SD	38.3 ± 7.1	23.8 ± 5.5	*p* < 0.001
Muscular area rectus anterior (MARA) (cm^2^)	Mean ± SD	4.97 ± 1.4	3.11 ± 0.89	*p* < 0.001
Muscular area index (MARAI) (cm^2^/m^2^)	Mean ± SD	1.73 ± 0.48	1.24± 0.34	*p* < 0.001
*X*-axis (cm)	Mean ± SD	3.96 ± 0.52	3.1 ± 0.41	*p* < 0.001
*Y*-axis (cm)	Mean ± SD	1.47 ± 0.33	1.16 ± 0.31	*p* < 0.001
Muscular circumference rectus (cm)	Mean ± SD	9.93 ± 1.26	7.73 ± 1.03	*p* < 0.001
Subcutaneous adipose tissue (SCAT) (cm)	Mean ± SD	0.55 ± 0.28	1.11 ± 0.38	*p* < 0.001

BMI: body mass index; SD: standard deviation; FFMI: fat-free mass index; BIA: Bioelectrical impedance analysis; DXA: dual-energy X-ray absorptiometry.

**Table 3 nutrients-14-03377-t003:** Correlations between ultrasound and other morphofunctional parameters.

	MARA (cm^2^)	MARAI (cm^2^/m^2^)	*X*-Axis (cm)	*Y*-Axis (cm)	Muscular Circumference (cm)	SCAT (cm)
BMI (kg/m^2^)	*r* = 0.385 *p* = 0.008	*r* = 0.339 *p* = 0.02	*r* = 0.183 *p* = 0.217	*r* = 0.380 *p* = 0.008	*r* = 0.280 *p* = 0.057	*r* = 0.330 *p* = 0.023
Fat-free mass (anthropometry) (kg)	*r* = 0.747 *p* < 0.001	*r* = 0.574 *p* < 0.001	*r* = 0.688 *p* < 0.001	*r* = 0.554 *p* < 0.001	*r* = 0.736 *p* < 0.001	*r* = −0.381 *p* = 0.009
FFMI (anthropometry) (kg/m^2^)	*r* = 0.712 *p* < 0.001	*r* = 0.642 *p* < 0.001	*r* = 0.605 *p* < 0.001	*r* = 0.568 *p* < 0.001	*r* = 0.659 *p* < 0.001	*r* = −0.286 *p* = 0.009
Fat-free mass (BIA) (kg)	*r* = 0.780 *p* < 0.001	*r* = 0.612 *p* < 0.001	*r* = 0.703 *p* < 0.001	*r* = 0.607 *p* < 0.001	*r* = 0.763 *p* < 0.001	*r* = −0.405 *p* = 0.006
FFMI (BIA) (kg/m^2^)	*r* = 0.774 *p* < 0.001	*r* = 0.710 *p* < 0.001	*r* = 0.635 *p* < 0.001	*r* = 0.660 *p* < 0.001	*r* = 0.714 *p* < 0.001	*r* = −0.325 *p* = 0.029
Phase angle (°)	*r* = 0.695 *p* < 0.001	*r* = 0.675 *p* < 0.001	*r* = 0.578 *p* < 0.001	*r* = 0.623 *p* < 0.001	*r* = 0.632 *p* < 0.001	*r* = −0.589 *p* < 0.001
Fat-free mass (DXA) (kg)	*r* = 0.670 *p* < 0.001	*r* = 0.505 *p* = 0.002	*r* = 0.480 *p* = 0.004	*r* = 0.616 *p* < 0.001	*r* = 0.677 *p* < 0.001	*r* = −0.570 *p* = 0.005
FFMI (DXA) (kg/m^2^)	*r* = 0.678 *p* < 0.001	*r* = 0.567 *p* < 0.001	*r* = 0.491 *p* = 0.003	*r* = 0.576 *p* < 0.001	*r* = 0.680 *p* < 0.001	*r* = −0.610 *p* = 0.002
Handgrip strength (kg)	*r* = 0.790 *p* < 0.001	*r* = 0.687 *p* < 0.001	*r* = 0.718 *p* < 0.001	*r* = 0.625 *p* < 0.001	*r* = 0.779 *p* < 0.001	*r* = −0.589 *p* < 0.001

MARA: muscular area rectus anterior; MARAI: muscular area index; SCAT: subcutaneous adipose tissue; BMI: body mass index; FFMI: fat-free mass index; BIA: bioelectrical impedance analysis; DXA: dual-energy X-ray absorptiometry.

**Table 4 nutrients-14-03377-t004:** Correlations between the ultrasound measurements and the different respiratory variables.

	Total Exacerbations	Severe Exacerbations	FEV1 (%)	FVC (%)	FEV1/FVC (%)
Muscular area rectus anterior (MARA) (cm^2^)	*r* = 0.019 *p* = 0.89	*r* = 0.161 *p* = 0.286	*r* = 0.445 *p* = 0.005	*r* = 0.376 *p* = 0.02	*r* = 0.344 *p* = 0.037
Muscular area index (MARAI) (cm^2^/m^2^)	*r* = 0.052 *p* = 0.730	*r* = 0.148 *p* = 0.326	*r* = 0.328 *p* = 0.044	*r* = 0.299 *p* = 0.068	*r* = 0.234 *p* = 0.164
*X*-axis (cm)	*r* = 0.019 *p* = 0.899	*r* = 0.092 *p* = 0.545	*r* = 0.279 *p* = 0.090	*r* = 0.270 *p* = 0.101	*r* = 0.107 *p* = 0.530
*Y*-axis (cm)	*r* = −0.057 *p* = 0.708	*r* = 0.038 *p* = 0.801	*r* = 0.444 *p* = 0.005	*r* = 0.398 *p* = 0.013	*r* = 0.350 *p* = 0.034
Muscular circumference rectus (cm)	*r* = −0.016 *p* = 0.917	*r* = 0.069 *p* = 0.646	*r* = 0.348 *p* = 0.032	*r* = 0.304 *p* = 0.064	*r* = 0.214 *p* = 0.203
Subcutaneous adipose tissue (SCAT) (cm)	*r* = 0.122 *p* = 0.420	*r* = 0.072 *p* = 0.634	*r* = −0.065 *p* = 0.698	*r* = −0.015 *p* = 0.931	*r* = −0.224 *p* = 0.183

FEV1: forced expiratory volume in 1 s; FVC: forced vital capacity.

**Table 5 nutrients-14-03377-t005:** Differences between muscle ultrasound parameters according to nutritional status.

		Normo-Nourished (*n* = 21)	Malnourished (*n* = 27)	*p*-Value
Muscular area rectus anterior (MARA) (cm^2^)	*m* ± SD	4.75 ± 1.65	3.37 ± 1.04	*p* = 0.014
Muscular area index (MARAI) (cm^2^/m^2^)	*m* ± SD	1.71 ± 0.51	1.28 ± 0.36	*p* = 0.016
*X*-axis (cm)	*m* ± SD	3.74 ± 0.65	3.37 ± 0.61	*p* = 0.279
*Y*-axis (cm)	*m* ± SD	1.45 ± 0.36	1.17 ± 0.28	*p* = 0.010
Muscular circumference rectus (cm)	*m* ± SD	9.48 ± 1.69	8.26 ± 1.34	*p* = 0.097
Subcutaneous adipose tissue (SCAT) (cm)	*m* ± SD	0.73 ± 0.37	0.95 ± 0.46	*p* = 0.872

*m*: mean; SD: standard deviation.

## Data Availability

Not applicable.

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
