# Peer review of "Usefulness of Muscle Ultrasonography in the Nutritional Assessment of Adult Patients with Cystic Fibrosis"

_nutrients, 2022, doi:10.3390/nu14163377_

Round 1
Reviewer 1 Report
Sánchez-Torralbo et al have performed an observational study whose purpose was to assess the usefulness of muscle ultrasonography in the diagnosis of malnutrition in patients with cystic fibrosis, comparing the results with other body composition techniques such as anthropometry, handgrip strength, BIA and DXA. The authors also assessed the association with the nutritional and respiratory status during patient follow-up. As the authors point out, this would be the first research in adults that assesses the use of muscle ultrasonography in adults. Souza [1] conducted a similar study in children. The authors conclude that in adults with cystic fibrosis, the measurements collected by muscle ultrasonography of the quadriceps rectus femoris correlate adequately with body composition techniques such as anthropometry
While their findings are interesting, several methodological and analytical issues are of concern:
Major comments:
1. The purpose of this research does not seem to be achieved with the results obtained. The authors perform the ROC analysis only for one of the measurements obtained by muscle ultrasonography (MARA), but we do not know the role played by the rest of the determinations in order to establish a diagnosis of malnutrition. Would the clinician use only the MARA to make a decision about the patient's malnutrition status? I suggest the authors obtain a score based on ultrasonographic variables. The score could be obtained by performing a multidimensional logistic analysis in which the dependent variable would be the presence or absence of malnutrition and the predictors would be the variables obtained by ultrasonography. In obtaining the model, a selection of variables (prospective, retrospective, best subset, etc.) should be performed. Obviously this score must take sex into account, because as Tables 2 and 3 of the manuscript reflect, the variables included in muscle ultrasonography are strongly sex-dependent.
2. The sample size is very small (24 males and 24 females), which does not allow for adequate data analysis. It is possible that the logistical analysis suggested in the previous point could be disappointing for this reason.
3. The manuscript seems disorderly and without a precise meaning. In this research there are two essential elements, namely: malnutrition in adult patients with cystic fibrosis and the markers for its diagnosis, being the purpose of the work to compare the authors' proposal based on ultrasonography in muscle with the rest of markers. From this point of view, it would be advisable to present a first table describing simultaneously the entire study population and each of the cohorts determined by the presence or absence of malnutrition. In this table, the general characteristics, respiratory status and of the patients could be summarized. Tables 2 and 3 should be summarized in a single table comparing variable anthropometric determinations potentially useful for the diagnosis of malnutrition. Here the population should be disaggregated by sex, but the entire population should not be described, as the results would be conditioned by the distribution of sexes (in this study there are as many males as females, but the results would be different if the percentages of males and females were different).
Minor comments
1. In lines 179 and 180, the authors say (sic) “Intraclass correlation coefficient was calculated between the different muscle ultrasound measurements”. Intraclass correlation coefficients are intended to measure the agreement of a set of quantitative assessments obtained with different measurement instruments or evaluators. Shrout [2] gives six different forms of intraclass coefficient (basically agreement and consistency). Could the authors clarify which intraclass correlation coefficient they have used? In no way could it be a measure of agreement.
2. In the description of the GLIM criteria, the authors use the operators "OR" "AND" in the form "and/or" (line 160). This expression is not academic, since, unless otherwise indicated, "OR" is an inclusive operator. The expression: "A or B occurs" unambiguously indicates that at least one of the two events occurs. I believe that the spurious operator "and/or" should simply be deleted, or if appropriate, replaced by "or".
3. It is possible that the criterion for the selection of the cut-off for the MARA described in line 191 is that of the point closest to the corner (0,1). In any case, it is expressed incorrectly.
4. The area under the ROC curve (AUC), sensitivity and specificity corresponding to the chosen cutoff should be given by a 95% confidence interval. Such intervals may be very long due to the small sample size (lines 272 and 273).
5. Authors should indicate the meaning of the acronym GLIM the first time it appears (line 28). [Global Leadership Initiative on Malnutrition].
Reference
[1] Souza, Rodrigo Pereira de, et al. "the use of ultrasonography to evaluate muscle thickness and subcutaneous fat in children and adolescents with cystic fibrosis." Revista Paulista de Pediatria 36 (2018): 457-465.
[2] Shrout, Patrick E., and Joseph L. Fleiss. "Intraclass correlations: uses in assessing rater reliability." Psychological bulletin 86.2 (1979): 420.
Author Response
Reviewer 1: Sánchez-Torralbo et al have performed an observational study whose purpose was to assess the usefulness of muscle ultrasonography in the diagnosis of malnutrition in patients with cystic fibrosis, comparing the results with other body composition techniques such as anthropometry, handgrip strength, BIA and DXA. The authors also assessed the association with the nutritional and respiratory status during patient follow-up. As the authors point out, this would be the first research in adults that assesses the use of muscle ultrasonography in adults. Souza [1] conducted a similar study in children. The authors conclude that in adults with cystic fibrosis, the measurements collected by muscle ultrasonography of the quadriceps rectus femoris correlate adequately with body composition techniques such as anthropometry
Authors:
Dear Reviewer,
Thank you for giving us the opportunity to improve our article “Usefulness of Muscle Ultrasonography in the Nutritional Assessment of Adult Patients with Cystic Fibrosis.”
The various suggestions have been incorporated into the new version wherever applicable. Please find below our responses and the action taken to all the suggestions and comments.
Please see the attachment to check the new version of the manuscript.
Once again, we very much appreciate all the work with the review.
Yours sincerely,
Dr. Francisco José Sánchez Torralvo
Dr. Gabriel Olveira
R:
While their findings are interesting, several methodological and analytical issues are of concern:
Major comments:
- The purpose of this research does not seem to be achieved with the results obtained. The authors perform the ROC analysis only for one of the measurements obtained by muscle ultrasonography (MARA), but we do not know the role played by the rest of the determinations in order to establish a diagnosis of malnutrition. Would the clinician use only the MARA to make a decision about the patient's malnutrition status? I suggest the authors obtain a score based on ultrasonographic variables. The score could be obtained by performing a multidimensional logistic analysis in which the dependent variable would be the presence or absence of malnutrition and the predictors would be the variables obtained by ultrasonography. In obtaining the model, a selection of variables (prospective, retrospective, best subset, etc.) should be performed. Obviously this score must take sex into account, because as Tables 2 and 3 of the manuscript reflect, the variables included in muscle ultrasonography are strongly sex-dependent.
A: Thank you for your appreciation.
We have used MARA to perform the ROC analysis because it is the measurement that, having a good correlation with other measures of fat-free mass, has most consistently shown its relationship with nutritional status in the analyzes carried out.
We have tried to perform a multidimensional logistic analysis, taking into account variables such as sex, but the limited sample size makes the analysis drastically lose power.
- The sample size is very small (24 males and 24 females), which does not allow for adequate data analysis. It is possible that the logistical analysis suggested in the previous point could be disappointing for this reason.
A: Yes, indeed, the short sample size does not allow for adequate data analysis that takes into account gender. The number of men and women in the sample is the same, so their weight should be offset. Something that we have verified is that both groups lose statistical significance equally when stratifying, which can allow us to think that one group does not have more weight than another in the global results.
- The manuscript seems disorderly and without a precise meaning. In this research there are two essential elements, namely: malnutrition in adult patients with cystic fibrosis and the markers for its diagnosis, being the purpose of the work to compare the authors' proposal based on ultrasonography in muscle with the rest of markers. From this point of view, it would be advisable to present a first table describing simultaneously the entire study population and each of the cohorts determined by the presence or absence of malnutrition. In this table, the general characteristics, respiratory status and of the patients could be summarized. Tables 2 and 3 should be summarized in a single table comparing variable anthropometric determinations potentially useful for the diagnosis of malnutrition. Here the population should be disaggregated by sex, but the entire population should not be described, as the results would be conditioned by the distribution of sexes (in this study there are as many males as females, but the results would be different if the percentages of males and females were different).
A: We very much appreciate your suggestion. We have changed table 1 with the suggestions made. We summarized tables 2 and 3 in a single table comparing anthropometric and morphofunctional determinations, disaggregating by sex.
Minor comments
- In lines 179 and 180, the authors say (sic) “Intraclass correlation coefficient was calculated between the different muscle ultrasound measurements”. Intraclass correlation coefficients are intended to measure the agreement of a set of quantitative assessments obtained with different measurement instruments or evaluators. Shrout [2] gives six different forms of intraclass coefficient (basically agreement and consistency). Could the authors clarify which intraclass correlation coefficient they have used? In no way could it be a measure of agreement.
A: Thank you for your appreciation. We have reviewed it and it does not make sense to apply a measure of agreement, as long as all the measurements have been made by the same observer. We have deleted the sentences about it.
- In the description of the GLIM criteria, the authors use the operators "OR" "AND" in the form "and/or" (line 160). This expression is not academic, since, unless otherwise indicated, "OR" is an inclusive operator. The expression: "A or B occurs" unambiguously indicates that at least one of the two events occurs. I believe that the spurious operator "and/or" should simply be deleted, or if appropriate, replaced by "or".
A: Thank you for your appreciation. We have made the suggested change.
- It is possible that the criterion for the selection of the cut-off for the MARA described in line 191 is that of the point closest to the corner (0,1). In any case, it is expressed incorrectly.
A: Thank you for your appreciation. We have improved the phrasing. The optimal cut-point was the point maximizing the product of sensitivity and specificity (Liu X. Classification accuracy and cut point selection. Statistics in Medicine. 2012;31(23):2676–2686. doi: 10.1002/sim.4509.)
- The area under the ROC curve (AUC), sensitivity and specificity corresponding to the chosen cutoff should be given by a 95% confidence interval. Such intervals may be very long due to the small sample size (lines 272 and 273).
A: We very much appreciate your suggestion. Unfortunately, our calculation tool used does not provide a confidence interval for the AUC, as occurred in another article already published in this journal: Nutrients 2022, 14, 1851. https://doi.org/10.3390/nu14091851
- Authors should indicate the meaning of the acronym GLIM the first time it appears (line 28). [Global Leadership Initiative on Malnutrition].
A: Thank you for your appreciation. We have made the suggested change both for the abstract and the text.
Reviewer 2 Report
This is a study with a double objective: to compare the results of muscle ultrasound with other body composition techniques and also to study their association with the nutritional and respiratory status.
The study is very well done, although there are some aspects to review.
In the abstract, "Y-axis" appears in the conclusions, which is not defined in the material and methods section of the abstract.
I understand that the Y-axis is what in other publications is the thickness of the muscle.
Fat-free mass (FFM) were estimated according to three methods:
1) formulas of Siri and Durnin from anthropometry
2) BIA (does not indicate the formula used)
3) DXA (does not indicate the formula used)
Regarding malnutrition criteria, they are very broad and could lead to overestimating its prevalence.
Regarding the phenotypic criteria, the three used do not have the same prevalence and prognostic importance:
1) more than 5% of unintentional weight loss in 6 months,
2) low BMI (BMI below 20 kg/m2),
3) decrease in muscle mass determined by DXA (gold standard)
Regarding etiological GLIM criteria, authors considered cystic fibrosis as an inflammatory condition. But in the inclusion criteria they indicated that they were in a situation of clinical stability. All patients were considered inflamed regardless of their situation.
Nearly half were diabetic. It would be interesting to see if there was any difference. The same comment regarding the existence of pancreatic insufficiency.
The correlations found between ultrasound measurements and FFMI measured by DXA (gold standard) are only moderate. The most intense are MARA and circumference.
It is noteworthy that the correlation is more intense between FFMI-DXA with transverse subcutaneous adipose tissue (SCAT) than with X axis of the muscle.
It is also noteworthy that correlation intensity is lost when MARAI is used.
It is also noteworthy that FFMI-DXA correlates better with X axis than with Y axis, but when comparing malnutrition or respiratory function only Y axis is significant.
All these contradictions are not commented on in the discussion.
The authors indicate that the correlation between the different intra-observer measurements was very high. But they don't show it in the results section.
In summary, it is a very well designed work and carried out with some small nuances.
Author Response
Reviewer 2:
This is a study with a double objective: to compare the results of muscle ultrasound with other body composition techniques and also to study their association with the nutritional and respiratory status.
The study is very well done, although there are some aspects to review.
Authors:
Dear Reviewer,
Thank you for giving us the opportunity to improve our article “Usefulness of Muscle Ultrasonography in the Nutritional Assessment of Adult Patients with Cystic Fibrosis.”
The various suggestions have been incorporated into the new version wherever applicable. Please find below our responses and the action taken to all the suggestions and comments.
Please see the attachment to check the new version of the manuscript.
Once again, we very much appreciate all the work with the review.
Yours sincerely,
Dr. Francisco José Sánchez Torralvo
Dr. Gabriel Olveira
R:
In the abstract, "Y-axis" appears in the conclusions, which is not defined in the material and methods section of the abstract. I understand that the Y-axis is what in other publications is the thickness of the muscle.
A: Thank you for your appreciation. Yes, indeed, it is the transverse muscle thickness, as explained in the material and methods section. We have not introduced the Y axis in the abstract due to text limit reasons, so we have removed it from the conclusions to avoid confusion.
Fat-free mass (FFM) were estimated according to three methods:
1) formulas of Siri and Durnin from anthropometry
2) BIA (does not indicate the formula used)
3) DXA (does not indicate the formula used)
A: Thank you for your appreciation. For BIA and DXA we used the formulas integrated in the software described in material and methods. We have improved the wording of the section.
Regarding malnutrition criteria, they are very broad and could lead to overestimating its prevalence.
Regarding the phenotypic criteria, the three used do not have the same prevalence and prognostic importance:
1) more than 5% of unintentional weight loss in 6 months,
2) low BMI (BMI below 20 kg/m2),
3) decrease in muscle mass determined by DXA (gold standard)
A: Precisely because of the difference between the prevalence of the different phenotypic criteria, we consider it important to know the prevalence of malnutrition according to GLIM criteria, which give great importance to muscle mass, which is more affected in these patients than other classic criteria such as BMI. It's something we discussed in the discussion section.
Regarding etiological GLIM criteria, authors considered cystic fibrosis as an inflammatory condition. But in the inclusion criteria they indicated that they were in a situation of clinical stability. All patients were considered inflamed regardless of their situation.
A: Thank you for your appreciation. We have revised the etiological criteria and included data on reduced assimilation (especially pancreatic insufficiency) and a variable of chronic inflammation (Glasgow Prognostic Score).
Nearly half were diabetic. It would be interesting to see if there was any difference. The same comment regarding the existence of pancreatic insufficiency.
A: Thank you for your appreciation. No significant differences were found in ultrasound measurements between patients with impaired carbohydrate metabolism or pancreatic insufficiency and those without.
The correlations found between ultrasound measurements and FFMI measured by DXA (gold standard) are only moderate. The most intense are MARA and circumference.
It is noteworthy that the correlation is more intense between FFMI-DXA with transverse subcutaneous adipose tissue (SCAT) than with X axis of the muscle.
It is also noteworthy that correlation intensity is lost when MARAI is used.
It is also noteworthy that FFMI-DXA correlates better with X axis than with Y axis, but when comparing malnutrition or respiratory function only Y axis is significant.
All these contradictions are not commented on in the discussion.
A: Thank you for your appreciation that improves our work.
In the discussion we note that previous works have pointed out differences in the estimation of fat-free mass between DXA and BIA and anthropometry in patients with CF, finding some overestimation in FFM using skinfold measurements and BIA. It is perhaps due to this overestimation that BIA and anthropometry measurements have a better correlation with purely muscular measurements such as MARA and muscle circumference. On the other hand, the formulas used for the calculation of FFM vary according to the technique
It is a fact that correlation intensity gets worst when MARAI is used with ALL techniques. This makes us think that the adjustment for height makes the correlation worse and that is why we do not use it in successive calculations. In any case, we have corrected some numerical mistakes detected in the table, including inversion of X and Y axis results for DXA.
The authors indicate that the correlation between the different intra-observer measurements was very high. But they don't show it in the results section.
A: Thank you for your appreciation. We have reviewed it at the request of another reviewer and it does not make sense to apply a measure of agreement, as long as all the measurements have been made by the same observer. We have deleted the sentences about it.
In summary, it is a very well designed work and carried out with some small nuances.